# Identification of Risk Factors for Sexual Dysfunction after Multimodal Therapy of Locally Advanced Rectal Cancer and Their Impact on Quality of Life: A Single-Center Trial

**DOI:** 10.3390/cancers14235796

**Published:** 2022-11-24

**Authors:** Peter Tschann, Markus Weigl, Thomas Brock, Jürgen Frick, Oliver Sturm, Jaroslav Presl, Tarkan Jäger, Michael Weitzendorfer, Philipp Schredl, Patrick Clemens, Helmut Eiter, Philipp Szeverinski, Christian Attenberger, Veronika Tschann, Walter Brunner, Alexander De Vries, Klaus Emmanuel, Ingmar Königsrainer

**Affiliations:** 1Department of General and Thoracic Surgery, Academic Teaching Hospital, 6800 Feldkirch, Austria; 2Department of Surgery, Paracelsus Medical University, 5020 Salzburg, Austria; 3Department of Radio-Oncology, Academic Teaching Hospital, 6800 Feldkirch, Austria; 4Institute of Medical Physics, Academic Teaching Hospital, 6800 Feldkirch, Austria; 5Private University in the Principality of Liechtenstein, 9495 Triesen, Liechtenstein; 6Department of Internal Medicine II, Academic Teaching Hospital, 6800 Feldkirch, Austria; 7Department of General, Visceral, Endocrine and Transplant Surgery, Cantonal Hospital of St. Gallen, 9000 St. Gallen, Switzerland

**Keywords:** rectal cancer, sexual dysfunction, quality of life, multimodal therapy

## Abstract

**Simple Summary:**

Sexual dysfunction (SD) after rectal cancer surgery is common, inadequately discussed, multifactorial and often untreated. It is well known that postoperative SD is caused by nerve damage or vascular injury of the vasa nervosa during surgery or perioperative therapy. The majority of the studies have focused on male dysfunction, missing female data or explored SD only as a secondary outcome. Moreover, the impact on quality of life is rarely described in previous literature. The aim of this study was to evaluate risk factors for worse sexual function and quality of life after multimodal therapy and surgery for locally advanced rectal cancer.

**Abstract:**

Purpose: Sexual function is crucial for the quality of life and can be highly affected by preoperative therapy and surgery. The aim of this study was to identify potential risk factors for poor sexual function and quality of life. Methods: Female patients were asked to complete the Female Sexual Function Index (FSFI-6). Male patients were demanded to answer the International Index of Erectile Function (IIEF-5). Results: In total, 79 patients filled in the questionary, yielding a response rate of 41.57%. The proportion of women was represented by 32.91%, and the median age was 76.0 years (66.0–81.0). Sexual dysfunction appeared in 88.46% of female patients. Severe erectile dysfunction occurred in 52.83% of male patients. Univariate analysis showed female patients (OR: 0.17, 95%CI: 0.05–0.64, *p* = 0.01), older age (OR: 0.34, 95%CI 0.11–1.01, *p* = 0.05), tumor localization under 6cm from the anal verge (OR: 4.43, 95%CI: 1.44–13.67, *p* = 0.01) and extension of operation (APR and ISR) (OR: 0.13, 95%CI: 0.03–0.59, *p* = 0.01) as significant risk factors for poor outcome. Female patients (OR: 0.12, 95%CI: 0.03–0.62, *p* = 0.01) and tumors below 6 cm from the anal verge (OR: 4.64, 95%CI: 1.18–18.29, *p* = 0.03) were shown to be independent risk factors for sexual dysfunction after multimodal therapy in the multivariate analysis. Quality of life was only affected in the case of extensive surgery (*p* = 0.02). Conclusion: Higher Age, female sex, distal tumors and extensive surgery (APR, ISR) are revealed risk factors for SD in this study. Quality of life was only affected in the case of APR or ISR.

## 1. Introduction

The management of rectal cancer has rapidly improved in the last 30 years. Much focus is placed on perioperative morbidity and oncological outcome. Multidisciplinary therapy including preoperative radiation therapy combined with or without chemotherapy and surgery represents the gold standard for advanced rectal cancer patients [1]. Due to improvement in survival, more attention could be turned to the quality of life and sexuality [2]. Sexual function is a crucial part of the quality of life and can be strongly affected by preoperative therapy and surgery [2,3]. Already in 1982, Heald et al. first described the importance of the surgical plane for an improved local recurrence rate [4]. Moreover, autonomic nerves started to gain more and more attention since that time. More recent data showed a significant advantage for nerve-sparing techniques in rectal cancer surgery regarding urinary and sexual function [5]. Several symptoms are known such as bladder dysfunction, low anterior resection syndrome (LARS) and sexual dysfunction (SD) to reduce the quality of life after rectal cancer surgery.

The majority of the studies have focused on male dysfunction, missing female data or explored SD only as a secondary outcome [2]. Only a few investigators have analyzed the difference between the type and technique of resection, differences in pre- and postoperative therapy and the outcome regarding sexual function and quality of life. However, SD after rectal cancer surgery is common, inadequately discussed, multifactorial and often untreated [6]. It is well known that postoperative SD is caused by nerve damage or vascular injury of the vasa nervosa during surgery and occurs between 11% and 55% of patients [7]. Furthermore, pelvic radiotherapy induces an inflammatory response and vascular and neurological damage with genital fibrosis in women [8]. It is well known that radiotherapy is a risk factor for poor sexual outcomes in both men and women [9]. Other proposed risk factors in rectal cancer patients include the type of surgery (abdominoperineal resection (APR), intersphincteric resection (ISR) and low anterior resection (LAR)), presence of stoma, older age and psychological factors [8].

The aim of this study was to evaluate and identify potential risk factors for the poor sexual function of locally advanced rectal cancer patients who underwent preoperative therapy followed by radical surgery and the impact on the quality of life.

## 2. Methods

### 2.1. Patients and Eligibility

All patients with locally advanced rectal cancer (cT3, cT4, N+) who received neoadjuvant therapy at the department of Radio-Oncology followed by resection in curative intention at the Department of Surgery between January 1998 and December 2020 were included in a prospective maintained population-based cohort database from the Academic Teaching Hospital in Feldkirch. All living patients were contacted via a telephone interview in accordance with ethical review guidelines (EK-0.04-413) and after institutional review board approval. If patients did not respond or were not available, they were excluded from this study. Erectile dysfunction in male patients is defined as the inability to achieve or maintain an erection sufficient for satisfactory sexual performance [10,11]. Male patients were asked to complete the International Index of Erectile Function (IIEF-5) [10,11], a 15-item validated tool containing five domains: erectile function, orgasmic function, sexual desire, intercourse satisfaction and overall satisfaction.Female sexual dysfunction traditionally includes disorders of desire and libido, arousal, pain and discomfort, and inhibited orgasm [12]. Female patients were asked to complete the Female Sexual Function Index (FSFI-6) [12], a 19-item validated tool containing six domains: desire, arousal, lubrication, orgasm, satisfaction and pain. Postoperative bowel function was assessed using the LARS score. Further health-related quality of life assessment was performed through scales including the European Organization for Research and Treatment of Cancer Quality of Life (EORTC-QLQ-C30) [13].

A colonoscopy including tissue biopsy and histological examination of the tumor was performed before therapy in all patients. Tumor height was defined with rectoscopy (anal verge), and local tumor staging was assessed with pelvic magnet resonance imaging (MRI) and in case of very low carcinomas or questions over sphincter involvement with endorectal ultrasound. A computer tomography (CT) procedure of the trunk was performed to rule out potential distant metastasis. The variables included in the study were: age, sex, body mass index (BMI), American Society of Anesthesiologists Score (ASA) [14], tumor location (in cm of the anal verge), preoperative chemoradiation therapy (CRT), comorbidities, preoperative carcinoembryonic antigen (CEA) level, clinical stages, surgical approach, protective defunctioning stoma, operative time, pre- and postoperative C-reactive protein (CRP) values, pre- and postoperative white blood cells (WBC) count, pre- and postoperative hemoglobin (Hb) value, postoperative pathological stage, duration of hospital stay, postoperative complications (according to the Clavien–Dindo classification [15]), local recurrence, distant metastasis, cancer-related death, disease-free survival (DFS) and overall survival (OS). Clinical and pathological staging was based on the 8^th^ edition of the Union for International Cancer Control (UICC) TNM Classification of malignant tumors [16].

### 2.2. Treatment Strategy

All rectal cancer cases were preoperatively discussed in a multidisciplinary team discussion and neoadjuvant therapy was indicated in accordance with international guidelines. Neoadjuvant therapy was performed in the case of locally advanced mid and low rectal cancer determined by CT scan of the trunk, pelvic MRI, colonoscopy and endorectal ultrasound either as combined chemoradiation therapy (long term) or as short-course radiotherapy.

### 2.3. Follow-Up

Follow-up was conducted as described in the international guidelines of the European Society of Medical Oncology and German S3 Leitlinie [1,17]. This occurred every three months during the first year after surgery, followed by a six-month clinical examination interval. CT and pelvic MRI scans were performed yearly. Colonoscopy was routinely carried out one year after the surgery and depending on the result the interval was determined.

### 2.4. Statistical Analysis

Statistical analysis was performed using SPSS^®^ (IBM™, New York, NY, USA). Continuous data of the patient characteristics were tested for normal distribution using the Shapiro–Wilk-Test [18]. Therefore, data were presented as the median and interquartile range (IQR). Continuous data of included and excluded patients were assessed by either the *t*-test or Mann–Whitney-U-test. Categorical data are presented in absolute numbers (percent) and were assessed using the Chi-square test or the exact Fisher test for small samples. Variables that could potentially influence sexual function were subjected to univariate analysis. The odds ratio (OR), its standard error and 95% confidence interval (CI) were calculated according to Altman, 1991 [19]. Parameters with a *p*-value ≤ 0.05 were entered in a multivariate logistic regression analysis. Data were collected using Excel^®^ (Microsoft™, Seattle, WA, USA). Significance was set at a *p*-value of <0.05.

## 3. Results

### 3.1. Patients’ Characteristics

A total number of 190 eligible patients who underwent rectal cancer surgery were identified from the database, of which 79 (41.57%) were contactable and responded to the questioning. Notably, 26 patients (32.91%) were female. The median age was 76.0 years (66.0–81.0), respectively. Median BMI was 25.0 (24.0–29.0), and most of the patients had a score of ASA II (60.27%). The tumors’ median inferior edge was 7.0 cm from the anal verge. In addition, 15 patients (18.99%) underwent previous abdominal surgery and 31 (39.24%) had one or more comorbidities. Most of the patients underwent long-term combined preoperative chemoradiation therapy (88.61%). Only nine patients had short-term therapy (11.39%). A LAR with primary anastomosis was performed in 56 patients (70.89%), an APR in 11 patients (13.92%) and an ISR with coloanal anastomosis was performed in 12 patients (15.91%), respectively. A protective defunctioning stoma was created in 49 patients (62.02%). The median operative time was 205.0 min (172.0–263.0). Complications were observed in 22 patients (28.21%). Anastomotic leakage occurred in 14 patients (17.72%). The defunctioning stoma was removed on average after 128 days (46.0–215.0). A permanent stoma was necessary for 26 patients (32.91%). The average duration of hospital stay was 14 days (12.0–20.0). Four patients (5.06%) developed a minor LARS and five patients (6.32%) had a major LARS, respectively. Quality of life (evaluated with the EORTC QLQ-C30) was high with a median value of 89.2 (82.78–94.9). Adjuvant chemotherapy was performed in 42 patients (53.16%). The median time between surgery and phone call was 73 months (50.0–173.0). A local recurrence was observed in three patients (3.8%) and distant metastasis occurred in eight patients (10.26%) during the follow-up. Patient characteristics are shown in Table 1.

### 3.2. Sexual Function

Most of the female patients (88.46%) reported a reduced sexual function in the telephone interview. Only three patients (11.54%) were satisfied with their sexual function. Those three underwent a LAR, no sexual satisfaction was reported in female patients in the case of ISR or APR, independent of the technique either minimally invasive surgery (MIS) or open resections. In male patients, severe erectile dysfunction was reported in 28 patients (52.83%). Hereby, moderate erectile dysfunction occurred in three (5.66%), mild to moderate in six (11.32%) and mild in five patients (9.43%). Only 11 patients (20.75%) reported no erectile dysfunction. In LAR with primary anastomosis 18 patients (43.90%) reported severe erectile dysfunction. In the case of APR or ISR, the majority of the patients had unsatisfying sexual function. If the resections were performed with minimally invasive surgery the sexual outcome was favorable compared to open surgery. The sexual outcome of the entire cohort and depending on the operative procedure is shown in Table 2.

Univariate risk factors’ analysis for poor sexual function indicated female patients (male vs. female: OR: 0.17, 95%CI: 0.05–0.64, *p* = 0.01), older age (<65 years vs. ≥65 years: OR: 0.34, 95%CI 0.11–1.01, *p* = 0.05), tumor localization under 6cm from the anal verge (<6 cm vs. ≥6 cm: OR: 4.43, 95%CI: 1.44–13.67, *p* = 0.01) and extension of operation (LAR vs. APR or ISR: OR: 0.13, 95%CI: 0.03–0.59, *p* = 0.01) as significant risk factors. Univariate analysis is shown in Table 3. Quality of life was significantly affected in the case of an extension of operation (*p =* 0.02). Other risk factors did not show a significant difference regarding the quality of life after multimodal therapy. Figure 1 shows the outcome of the EORTC QLQ-C30 in relation to the risk factors of the univariate analysis.

Sex, age, tumor localization from the anal verge and extension of operation were included in a multivariate analysis model. Female patients (male vs. female: OR: 0.12, 95%CI: 0.03–0.62, *p* = 0.01) and tumors below 6 cm from the anal verge (<6 cm vs. ≥6 cm: OR: 4.64, 95%CI: 1.18–18.29, *p* = 0.03) were shown to be significant risk factors for poor sexual function after therapy. Multivariate analysis is shown in Table 4.

## 4. Discussion

Numerous studies exist about the quality of life, bowel function, cancer-specific survival and morbidity after low anterior resection in the case of rectal cancer. Sexual function and intimacy are considered to be important aspects regarding the quality of life and should be key elements in determining outcomes of multidisciplinary interventions in rectal cancer patients [20]. In this study, we could reach at least 42% of the patients who received preoperative and surgical treatment for rectal cancer at our department. The overall rate of SD, defined as severe or moderate score in men (IIEF-5) and ≤19 score points in women (FSFI-6), was 68.35% (male: 58.49%, women: 88.46%), respectively. Moreover, these data showed female sex, tumor localization under 6cm from the anal verge, APR or ISR and open surgery as significant risk factors for poor sexual function in the univariate analysis and female sex as well as low tumors as independent risk factors in the multivariate analysis.

While multimodal therapy for rectal cancer has clearly evoked beneficial outcomes, patients are still negatively affected by adverse events of surgery, radio- and chemotherapy [21]. SD is common after rectal cancer treatment. Moreover, it is not common for patients to raise this issue because they are either embarrassed or do not see their symptoms related to the treatment [20,22]. SD may have a major impact on the psychological, emotional and social function of patients, especially in younger patients. This study was designed in accordance with previous literature using validated and standardized questionary tools to quantify sexual function after rectal cancer therapy.

Concordant with previously published studies in the literature, these data revealed a negative impact of therapy on both genders. The rate of SD was higher in female patients (88.46% vs. 58.49%). Furthermore, the female sex was a significant risk factor in the univariate (male vs. female: OR: 0.17, 95%CI: 0.05–0.64, *p* = 0.01) as well as in the multivariate analysis (male vs. female: OR: 0.14, 95%CI: 0.03–0.66, *p* = 0.01). Most of the studies focused on male erectile dysfunction, and research on female rectal cancer survivors’ sexual outcomes is limited [23]. Two main factors may explain the relatively high rates of SD in female patients: Firstly, early menopause following (neo)adjuvant treatment may lead to SD [23]. Moreover, vaginitis, vaginal atrophy and vaginal stenosis are described after radiotherapy of pelvic cancers [24]. The cause for radiotherapy-associated problems is multifactorial and involves vascular and genital tissue fibrosis, neurological damage, vaginal narrowing and loss of elasticity [7]. In addition, nerve injury and postoperative scarring around the vaginal region may contribute to dissatisfaction or avoidance of sexual intercourse [25]. Secondly, women are more likely than men to give up sexual activity after colorectal cancer therapy [23,26]. Furthermore, too little attention is paid in the follow-up discussions to the mediation of preventive measures such as vaginal dilation. However, in this study, data for SD in women are limited and should be interpreted with caution because of the relatively low number of included female patients. Moreover, half of the female patients were >75 years, where problems with sexual desire are known to be more common [27].

In the previous literature, different cutoff values of age were used to evaluate SD after rectal cancer treatment. SD is reported to increase with the age [7,28,29,30], which may be due to an increased rate of comorbidities on the one hand and reduced sexual desire in elderly patients on the other hand. Schmidt et al. [31] were able to show that younger rectal cancer patients suffer more from the pressure of a possible SD than older patients, thus, quality of life is more negatively affected in those patients. However, in accordance with previously published studies in the literature, these data could confirm age as a significant risk factor for poor sexual outcome in the univariate analysis (<65 years vs. ≥65 years: OR: 0.34, 95%CI 0.11–1.01, *p* = 0.05).

Tumor below 6 cm from the anal verge was shown to be another risk factor for poor sexual outcome in this study in the univariate- (<6 cm vs. ≥6 cm: OR: 4.43, 95%CI: 1.44–13.67, *p* = 0.01) as well as in the multivariate analysis (<6 cm vs. ≥6 cm: OR: 4.64, 95%CI: 1.18–18.29, *p* = 0.03) which is consistent with previously published studies [7,25]. The risk for pelvic nerve injuries is reported to increase in cancer localization under the peritoneal reflection [25]. Limited view and narrow pelvis in this area especially in male patients may lead to increased rates of nerve damage. The use of advanced sphincter-sparing techniques, including ISR and coloanal anastomosis may cause further irritation of sexual function due to either postoperative fibrosis or scarring alteration of the perineum and distal vaginal region [7].

The data in this study showed an inferior outcome of extensive surgery (APR or ISR) compared with patients undergoing a LAR with primary anastomosis in the univariate (LAR vs. APR or ISR: OR: 0.13, 95%CI: 0.03–0.59, *p* = 0.01). APR, ISR and Hartmann procedures are well-known risks to be factors for postoperative SD [7,32]. On the one hand, nerve damage during perineal dissection may cause more functional problems than in LAR. On the other hand, psychological factors associated with a stoma and parastomal hernia or a postoperative worse bowel function in the case of ISR could explain the higher rate of SD in those patients who underwent APR [7].

In accordance with some previously performed systematic reviews, the data in this study displayed no significant difference between MIS-performed procedures and open surgery regarding sexual function [33,34]. However, the magnified view which is offered in laparoscopic surgery leads to a more precise dissection of pelvic nerves and this could lead to a preserved sexual function.

These data showed that SD did not affect the quality of life significantly. The majority of the patients had an acceptable quality of life in the EORTC QLQ-C30-questionary with a median value of 89.2 (IQR: 82.78–94.9) independent of sexual function. Only 26.67% of the patients had the impression that the operative procedure had changed their sexual behavior. One reason that may explain this outcome is that only half of the patients considered sexual function important for the quality of life. Bowel function and the oncological outcome were more important than sexual function for the majority of the study cohort. Another interesting result of this study is a relatively low number of patients with LARS symptoms (*n* = 9, 13.23%). These data should be interpreted with caution. Current studies in different countries have found the prevalence of LARS in patients after sphincter-preserving surgery ranges between 30% and 80% [35,36,37,38]. A Meta-analysis by Ye et al. described a prevalence of 49.7% in a total of 5102 included patients. One reason for the low number of patients in this study is that the questionary was asked via telephone interview which implicates a selection bias and potentially uncomfortable questions were not answered completely correctly. Secondly, no baseline of sphincter functions is available which also implicates a selection bias.

This study has several limitations to be mentioned: The study was of a retrospective design. All patients were reached via phone call after completing therapy during or after the follow-up time. Information about the function before therapy is weak. Additionally, the response rate is below fifty percent. Both points represent indirectly a risk for selection bias. Furthermore, the average age in this cohort was high, and the time between therapy and the questionary is long in some cases which could cause a bias. Sexual function in the elderly is known to be less present than in younger age. Finally, SD is a multifactorial process, many confounding factors such as surgeons’ experience and partner-associated factors could cause a bias.

However, these data clearly showed age, female sex, extensive surgery (APR, ISR) and open surgery as significant risk factors for SD. Nonetheless, SD remains a serious problem with a high incidence in both, females and males after rectal cancer treatment. Nerve-sparing procedures and minimally invasive techniques could make a difference in experienced hands [5]. This study adds knowledge to the growing interest in SD after rectal cancer treatment. Every surgeon should be aware of SD and discuss the impact of therapy on sexual function preoperatively and ensure adequate consent and appropriate postoperative management strategies [2,39].

## 5. Conclusions

SD seems to be an underestimated complication after multimodal therapy for rectal cancer. Higher Age, female sex, extensive surgery (APR, ISR), and open surgery are revealed risk factors for SD in this study. Discussion about the impact of therapy on sexual function should be held preoperatively. More attention should be paid to nerve-sparing techniques and minimally invasive procedures as well as consequent postoperative support. Postoperative consultations should include a questionary about sexual function to improve management strategies in case of SD and reduction in quality of life. However, quality of life was not significantly affected in most patients who participated in this study independent of sexual function.

## Figures and Tables

**Figure 1 cancers-14-05796-f001:**
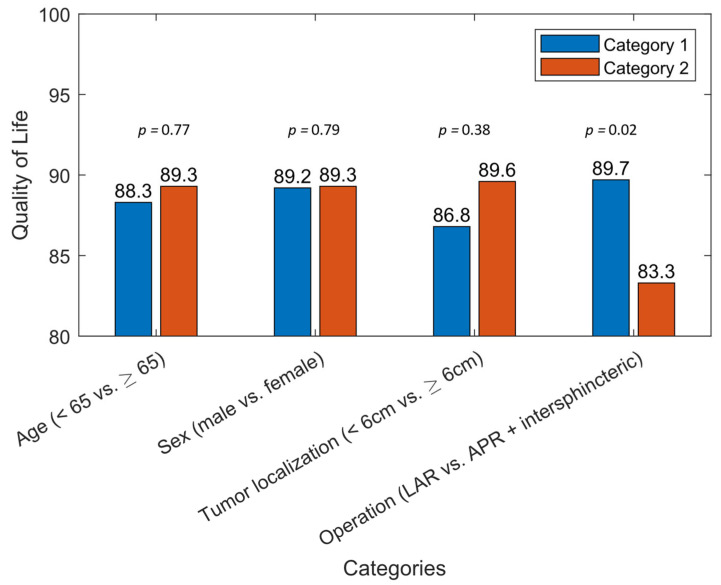
Results of the EORTC QLQ-C30 in relation to risk factors in the univariate analysis for sexual dysfunction after multimodal therapy of rectal cancer.

**Table 1 cancers-14-05796-t001:** Patient characteristics perioperative findings of included Patients.

Variables	*n =* 79
Sex, Male/Female, *n* (%)	53 (67.09%)/26 (32.91%)
Age, median (IQR) (y)	76.0 (66.0–81.0)
BMI, median (IQR) (kg/m^2^)	25.0 (24.0–29.0)
ASA Classification, *n* (%)	
I	8 (10.96%)
II	44 (60.27%)
III	21 (28.77%)
Tumor localization (in cm from the anal verge), median (IQR)	7.0 (4.0–10.0)
CEA level preoperative, median (IQR) (µg/l)	3.0 (2.0–4.0)
Previous abdominal surgery, *n* (%)	15 (18.99%)
Comorbidities, *n* (%)	31 (39.24%)
Coronary disease	5 (6.33%)
Pulmonary insufficiency	1 (1.27%)
Adipositas	10 (12.66%)
Kidney disease	4 (5.06%)
Hypertension	27 (34.18%)
Diabetes	2 (2.53%)
others	1 (1.27%)
Preoperative therapy, *n* (%)	
Short-term (5 × 5 Gy)	9 (11.39%)
Combined long-term chemo-radiation	70 (88.61%)
Operative method, *n* (%)	
Low anterior resection (LAR)	56 (70.89%)
Abdominoperineal resection (APR)	11 (13.92%)
Intersphincteric resection (ISR)	12 (15.19%)
Anastomosis, *n* (%) (total: *n* = 68)	
E-E	42 (53.16%)
S-E	15 (18.98%)
Pouch	11 (13.92%)
Protective defunctioning stoma, *n* (%)	49 (62.02%)
Operation time, min, median (IQR)	205.0 (172.0–263.0)
Hb preoperative, median (IQR) (g/dL)	134.0 (124.0–147.0)
Hb postoperative, median (IQR) (g/dL)	110.0 (99.0–119.0)
WBC postoperative, median (IQR) (G/L)	7.0 (6.0–9.0)
CRP postoperative day 1, median (IQR) (mg/dL)	8.0 (4.0–14.0)
Peak CRP within 14 days after the operation, median (IQR) (mg/dL)	11.0 (6.0–20.0)
Complications, *n* (%)	22 (28.21%)
Anastomotic leakage	14 (17.72%)
Wound infection	7 (8.86%)
Bowel obstruction	4 (5.06%)
Urinary dysfunction	1 (1.27%)
Clavien–Dindo Classification, *n* (%)	
I	5 (6.41%)
II	4 (5.13%)
III	16 (20.51%)
IV	1 (1.28%)
Stoma Reversal Time, median (IQR) (d)	128.0 (46.0–215.0)
Permanent Stoma, *n* (%)	26 (32.91%)
Duration of hospital stay, median (IQR) (d)	14.0 (12.0–20.0)
LARS-score, median (IQR) *	9.0 (0.0–15.0)
No LARS, *n* (%)	46 (67.64%)
Minor LARS, *n* (%)	4 (5.88%)
Major LARS, *n* (%)	5 (7.35%)
n/a, *n* (%)	13 (19.11%)
Quality of life (EORTC QLQ-C30), median (IQR)	89.2 (82.78–94.9)
Pathological yT Stage, *n* (%)	
Tis	15 (18.99%)
T1	2 (2.53%)
T2	31 (39.24%)
T3	28 (35.44%)
T4	3 (3.8%)
Pathological yN Stage, *n* (%)	
N0	57 (72.15%)
N1	16 (20.25%)
N2	6 (7.59%)
Postoperative UICC Stage, *n* (%)	
0	15 (18.98%)
I	24 (26.66%)
II	15 (18.98%)
III	21 (26.58%)
IV	4 (5.06%)
Grade of regression (Rödel et al.)	
1	3 (8.11%)
2	8 (21.62%)
3	20 (54.05%)
4	6 (16.22%)
n/a	42 (53.16%)
Adjuvant chemotherapy, *n* (%)	42 (53.16%)
Time between surgery and questionary, median (IQR), (months)	73.0 (50.0–173.0)
Local recurrence, *n* (%)	3 (3.8%)
Distant metastasis, *n* (%)	8 (10.26%)

Abbreviations: BMI = body mass index, ASA = American Society of Anesthesiologists, CEA = Carcinoembryonic antigen, UICC = Union for International Cancer Control, LAR = low anterior resection, APR = Abdominoperineal resection, ISR = Intersphincteric resection, E-E = end to end, S-E = side to end, Hb = Hemoglobin, IQR = Interquartile range, CRP = C-reactive protein, Gy = Gray. *: Patients who underwent APR were excluded from the LARS questionary. Values are given as median and IQR (interquartile range) or numbers and percentages.

**Table 2 cancers-14-05796-t002:** Sexual dysfunction of entire cohort and depending on operation procedure or technique, either MIS or open surgery and distance from the anal verge.

	Total	Type of Surgery			Operation Technique		Distance from the Anal Verge	
		LAR	APR	ISR	MIS	Open	<6 cm	≥6 cm
**Total number of male patients (IIEF 5):**	53	41	5	7	25	28	15	38
severe (1–7)	28 (52.83%)	18 (43.90%)	4 (80.0%)	5 (71.42%)	11 (44.0%)	17 (60.71%)	11 (50.00%)	17 (29.82%)
moderate (8–11)	3 (5.66%)	3 (7.31%)	0 (0.0%)	0 (0.0%)	2 (8.0%)	1 (3.57%)	0 (0.0%)	3 (5.26%)
mild to moderate (12–16)	6 (11.32%)	5 (12.91%)	0 (0.0%)	1 (14.28%)	3 (12.0%)	3 (10.71%)	0 (0.0%)	6 (10.52%)
mild (17–21)	5 (9.43%)	5 (12.91%)	0 (0.0%)	0 (0.0%)	2 (8.0%)	3 (10.71%)	1 (4.54%)	4 (7.01%)
no (22–25)	11 (20.75%)	10 (24.39%)	1 (20.0%)	1 (14.28%)	7 (28.0%)	4 (14.28%)	3 (13.63%)	8 (14.03%)
**Total number of female patients (FSFI 6):**	26	15	6	5	13	13	7	19
FSD (≤19)	23 (88.46%)	12 (80.0%)	6 (100%)	5 (100%)	10 (76.92%)	13 (100%)	7 (100%)	16 (84.21%)
no FSD (>19)	3 (11.54%)	3 (20.0%)	0 (0.0%)	0 (0.0%)	3 (23.07%)	0 (0.0%)	(0.0%)	3 (15.78%)

Abbreviations: IIEF = International Index of Erectile Dysfunction, FSFI = Female Sexual Function Index, LAR = Low anterior resection, APR = Abdominoperineal resection, ISR = Intersphincteric resection, MIS = minimally invasive surgery, FSD = Female sexual dysfunction. Values are given as numbers and percentages.

**Table 3 cancers-14-05796-t003:** Univariate analysis of potential risk factors for sexual dysfunction after rectal cancer therapy.

Variables	OR (95%CI)	*p*-Value
Sex (male vs. female)	0.17 (0.05–0.64)	0.01
BMI (kg/m^2^) (<25 vs. ≥25)	0.97 (0.35–2.71)	0.96
Age (<65 vs. ≥65)	0.34 (0.11–1.01)	0.05
ASA (I vs. II + III)	0.38 (0.09–1.7)	0.21
Comorbidities (yes vs. no)	2.25 (0.81–6.24)	0.12
Tumor localization (from the anal verge) (<6 cm vs. ≥6 cm)	4.43 (1.44–13.67)	0.01
UICC Stage preoperative (0–2 vs. 3–4)	0.66 (0.21–2.1)	0.48
UICC Stage postoperative (0–2 vs. 3–4)	1.22 (0.45–3.32)	0.69
pT Stage (T0+2 vs. T3+4)	0.75 (0.28–1.98)	0.56
pN Stage (N0 vs. N+)	1.63 (0.59–4.53)	0.35
pM Stage (M0 vs. M1)	0.67 (0.07–6.74)	0.73
CEA level (ng/mL) (<3 vs. ≥3)	2.22 (0.66–7.48)	0.2
Operation (LAR vs. APR/ISR)	0.13 (0.03–0.59)	0.01
Operation technique (MIS vs. open)	0.44 (0.17–1.16)	0.1
Complication (yes vs. no)	1.02 (0.35–2.92)	0.98
Local recurrence (yes vs. no)	0.98 (0.08–11.33)	0.99
Distant metastasis (yes vs. no)	0.82 (0.18–3.71)	0.79
Anastomotic leakage (yes vs. no)	0.59 (0.18–1.93)	0.39

Abbreviations: BMI = body mass index, ASA = American Society of Anesthesiologists, CEA = Carcinoembryonic antigen, UICC = Union for International Cancer Control, OR = Odds ratio, CI = Confidence interval, LAR = Low anterior resection, APR = Abdominoperineal resection, ISR = Intersphincteric resection, MIS = Minimally invasive surgery.

**Table 4 cancers-14-05796-t004:** Multivariate analysis of risk factors with a *p*-value <0.05 in the univariate analysis.

Variables	OR (95%CI)	*p*-Value
Sex (male vs. female)	0.12 (0.03–0.62)	0.01
Age (<65 vs. ≥65)	0.37 (0.82–1.68)	0.20
Tumor localization (from the anal verge) (<6 cm vs. ≥6 cm)	4.64 (1.18–18.29)	0.03
Operation (LAR vs. APR/ISR)	0.12 (0.01–1.09)	0.06
Operation (MIS vs. open)	0.3 (0.09–1.11)	0.07

Abbreviations: LAR = Low anterior resection, APR = abdominoperineal resection, ISR, Intersphincteric resection, MIS = Minimally invasive surgery, OR = Odds ratio, CI = Confidence interval.

## Data Availability

The datasets generated during and/or analyzed during the current study are available from the corresponding author upon reasonable request.

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
