# Peer review of "Identification of Risk Factors for Sexual Dysfunction after Multimodal Therapy of Locally Advanced Rectal Cancer and Their Impact on Quality of Life: A Single-Center Trial"

_cancers, 2022, doi:10.3390/cancers14235796_

Round 1

Reviewer 1 Report

This is an interesting study that adds to the relative paucity of information regarding postoperative female sexual dysfunction in this high-risk group of rectal cancer patients.

There are several issues that need to be resolved:

1) Definition of what constitutes severe sexual dysfunction is only given in the discussion (line 219 ff.). This belongs with the methods section. 

2) In Table 1 LARS subgroups and respective percentages should only be shown for patients with anastomosis. Likewise, for types of anastomosis, the denominator should only be 68 and not the total of 79 patients. Moreover, the clarity of the table would be greatly enhanced by printing the denominators for the respective calculations where they are different from the total group and clear structuring of sections (e.g., type of anastomosis (n=68): E-E, etc.)

3) According to literature, on average around 40% of patients with LAR would be expected to suffer from major LARS. How do the authors explain the exceptionally low incidence (9%) in their series, especially in view of the reported rather high leak rate of 18% that would also be expected to negatively impact function?

4) height of tumor from anal verge, type of surgery, and type of access are clearly interdependent, but it is quite difficult from Table 2 to understand the relationships. A separate table would be beneficial.

5) There are two instances, where the description of the effect and the given odds ratios do not match (Table 3: "operation technique"; Table 4: "Sex").

With an OR of 0.02, if MIS protects against severe SD, then the correct description in text and table would be "Operation technique (MIS vs. open)"(line 189).

Is the given value for the OR of 0.02 correct? 

With an OR of 0.12, if females have a higher risk for severe SD (line 206), then the correct text in the table should also be: "Sex (male vs. female)".

An alternative could be to recalculate the uni- and multivariate analyses and consistently show the increased ORs, so that they match the text.

Author Response

Dear Reviewer!

Thank you very much for reviewing this manuscript and for all your recommendations. We carefully reviewed the manuscript according to your suggestions and improvements. All corrections are marked and highlighted. I hope we answered all your queries correctly to improve this manuscript. Please don’t hesitate to contact me if you have further questions/suggestions.

Kind regards

Peter Tschann

Point for point answer:

This is an interesting study that adds to the relative paucity of information regarding postoperative female sexual dysfunction in this high-risk group of rectal cancer patients.

There are several issues that need to be resolved:

1) Definition of what constitutes severe sexual dysfunction is only given in the discussion (line 219 ff.). This belongs with the methods section. 

Added into the methods section. L87, L90

2) In Table 1 LARS subgroups and respective percentages should only be shown for patients with anastomosis. Likewise, for types of anastomosis, the denominator should only be 68 and not the total of 79 patients. Moreover, the clarity of the table would be greatly enhanced by printing the denominators for the respective calculations where they are different from the total group and clear structuring of sections (e.g., type of anastomosis (n=68): E-E, etc.)

Corrected and highlighted in table 1.

3) According to literature, on average around 40% of patients with LAR would be expected to suffer from major LARS. How do the authors explain the exceptionally low incidence (9%) in their series, especially in view of the reported rather high leak rate of 18% that would also be expected to negatively impact function?

You are right. We added this into the discussion section. L266-L273

4) height of tumor from anal verge, type of surgery, and type of access are clearly interdependent, but it is quite difficult from Table 2 to understand the relationships. A separate table would be beneficial.

You are right. We tried to clarify and added distance from the anal verge into table 2.

5) There are two instances, where the description of the effect and the given odds ratios do not match (Table 3: "operation technique"; Table 4: "Sex").

You are right. Corrected.

With an OR of 0.02, if MIS protects against severe SD, then the correct description in text and table would be "Operation technique (MIS vs. open)"(line 189).

Corrected

Is the given value for the OR of 0.02 correct? 

We carefully reviewed the univariate analysis. OR of 0.02 in MIS vs. open was not correct. We are sorry for that mistake and corrected it in the text and table 3 and deleted this category in table 4. In addition, Figure 1 was corrected.

With an OR of 0.12, if females have a higher risk for severe SD (line 206), then the correct text in the table should also be: "Sex (male vs. female)".

You are right. Corrected

An alternative could be to recalculate the uni- and multivariate analyses and consistently show the increased ORs, so that they match the text.

We carefully reviewed the univariate analysis. The given OR’s are correct and quoted correctly in the text.

Reviewer 2 Report

The limitations of the study make it impossible to draw conclusions.

The absence of pre-treatment data (FSFI-6, IIEF-5 and EORTC-QLQ-C30 tests) make it impossible to determine its changes after treatment of rectal cancer.

The use of the telephone survey adds selection bias and the lack of response of more than half of the patients included further increases the selection bias.

Much data is repeated in the text and in the tables.

The variables to include in a multivariate analysis have not been correctly selected.

The number of DS in women is striking. A previous hormonal analysis could have better oriented this question.

Author Response

Dear Reviewer!

Thank you very much for reviewing this manuscript and for all your recommendations. We carefully reviewed the manuscript. All corrections are marked and highlighted.

SD and quality of life after rectal cancer surgery is very much overlooked in the clinical day life. A randomized controlled trial is needed for a better understanding of functional problems after rectal cancer surgery. And yes, you are right this study has several limitations like mentioned in the text. But the majority of the studies are of a retrospective design with and all of them are limited by the design itself. Nevertheless, because of lack of randomized controlled trials regarding SD retrospective trials are important to have any information about SD after rectal cancer surgery.

Kind regard

Peter Tschann

Reviewer 3 Report

Important subject, not well studied in general literature. Well studied and presented.

Author Response

Dear Reviewer!

Thank you very much for reviewing this manuscript and for all your recommendations. We carefully reviewed the manuscript. All corrections are marked and highlighted. Please don’t hesitate to contact me if you have further questions/suggestions.

Kind regard

Peter Tschann